# An *Ex-Vivo* Culture System of Ovarian Cancer Faithfully Recapitulating the Pathological Features of Primary Tumors

**DOI:** 10.3390/cells8070644

**Published:** 2019-06-26

**Authors:** Farhana Ishrat Ghani, Kasumi Dendo, Reiko Watanabe, Kenji Yamada, Yuki Yoshimatsu, Takashi Yugawa, Tomomi Nakahara, Katsuyuki Tanaka, Hiroshi Yoshida, Masayuki Yoshida, Mitsuya Ishikawa, Naoki Goshima, Tomoyasu Kato, Tohru Kiyono

**Affiliations:** 1Division of Carcinogenesis and Cancer Prevention, National Cancer Center Research Institute, Tokyo 104-0045, Japan; 2Department of Cell Culture Technology, National Cancer Center Research Institute, Tokyo 104-0045, Japan; 3Pathology Division, National Cancer Center Hospital, Tokyo 104-0045, Japan; 4Department of Gynecology, National Cancer Center Hospital, Tokyo 104-0045, Japan; 5Molecular Profiling Research Center for Drug Discovery, National Institute of Advanced Industrial Science and Technology, 2-4-7 Aomi, Koto-ku, Tokyo 135-0064, Japan

**Keywords:** ovarian cancer, cell culture, xenograft

## Abstract

The success rate of establishing human cancer cell lines is not satisfactory and the established cell lines often do not preserve the molecular and histological features of the original tissues. In this study, we developed a novel culture method which can support proliferation of almost all primary epithelial ovarian cancer cells, as well as primary normal human oviductal epithelial cells. Cancer cells from fresh or frozen specimens were enriched by the anti-EpCAM antibody-conjugated magnetic beads, plated on Matrigel-coated plate and cultivated under the optimized culture conditions. Seventeen newly established ovarian cancer cell lines, which included all four major histotypes of ovarian cancer, were confirmed to express histotype-specific markers *in vitro*. Some of the cell lines from all the four histotypes, except mucinous type, generated tumors in immune-deficient mice and the xenograft tumor tissues recapitulated the corresponding original tissues faithfully. Furthermore, with poorly tumorigenic cell lines including mucinous type, we developed a novel xenograft model which could reconstruct the original tissue architecture through forced expression of a set of oncogenes followed by its silencing. With combination of the novel culture method and cell-derived xenograft system, virtually every epithelial ovarian cancer can be reconstituted in mice in a timely fashion.

## 1. Introduction

Although cancer cell growth cannot be controlled inside the body, it is often difficult to culture in vitro and the success rate is poor in most cases [1,2]. Moreover the reliability of commonly used cancer cell lines has recently been considered problematic. Among NCI60 cell lines, only 34 of 60 cell lines displayed the highest tissue similarity index compared to their tumors of origin and seven cell lines were identified as being of another origin than the originally presumed one [3]. Among 47 ovarian cancer cell lines, some of the cell lines did not resemble cognate tumor profiles at all [4]. It is also reported that many cell lines have acquired mutations not present in the original tumor DNA. Indeed, most popular cancer cell line, HeLa, can continuously acquire novel mutations during passages, indicating strong selective pressure for cancer cells under ordinary culture conditions [5]. There are three methods of generating an unlimited culture system of primary cancer tissue. Patient- derived xenograft (PDX), 3D culture and 2D or monolayer culture system. PDX is both time and money consuming and laborious as well. 3D culture is comparatively less time- and money-consuming than xenografts, but it is not such an easy and fast culture system [6]. Therefore, an efficient monolayer primary cell culture system that retains the molecular and histological features of primary tumor of each patient faithfully, could be the first choice of modeling cancer in vitro, which could be a source of information to predict novel therapeutic approaches for personalized medicine. The epithelial malignant ovarian tumors are classified into different histological types as follows: serous, mucinous (MC), endometrioid (EM), clear cell (CC), malignant Brenner tumors, carcinosarcoma, mixed epithelial tumor, undifferentiated carcinoma, and others [7].

Among these, serous, MC, EM and CC carcinomas are more frequently observed and comprise the major histological types. Serous type is further classified into low-grade and high-grade serous carcinomas (HGSC). Nowadays, ovarian cancer is considered to be not a disease of the ovaries as there is a growing body of evidence showing that most of the major histotypes originate from cells in parts of the reproductive organs other than the ovaries, such as the fallopian tubes or uterus [8] though the origin of MC type is still unknown. So far there have been several approaches to generate cell lines from primary ovarian cancers, but the success rates are not satisfactory. In one approach [9] the success rate was only twelve percent and all successful cell lines were established from ascites fluid, not from solid tumors. They were also from high-grade poorly differentiated cancers with serous (n = 4) not otherwise specified (n = 4) or mixed Müllerian (n = 1) histology and attempts for EM and CC type were unsuccessful. Though a recent report about characterization of twenty five ovarian cancer lines [10] demonstrates that the success rate is ninety five percent, only around half of the lines were established directly from the primary tissue and the rest were from ascites or xenografts. Also there are mouse models that do not represent the original tumor characteristics properly [11].

Here we report a robust and efficient culture method to generate monolayer cultures of primary ovarian cancer cells. We established and characterized 17 novel ovarian cancer cell lines, including eight cell lines out of eight primary tumor specimens consecutively obtained under the most optimized conditions. It took about one and a half months or less to establish each cell line. Isolation of selected cancer cells from the tumor tissue, Matrigel-coated plate and optimized medium enabled highly efficient establishment of ovarian cancer cell lines. We confirmed the characteristic protein expression in each histotype which are commonly seen in its ovarian cancer histology. As half of the representative cell lines transplanted into immune-deficient mice failed to establish xenograft tumors, we developed a method to generate xenograft tumors from those which previously failed to generate xenograft tumors, including mucinous cell lines by conditionally expressing a set of ectopic oncogenes followed by its silencing. All the xenograft tissues thus established faithfully recapitulated the original histology as well as the marker expression patterns. The proposed fast and robust method to establish cancer cell lines and xenografts from ovarian cancer specimens will provide crucial information for therapeutic approaches in personalized treatment of ovarian cancers.

## 2. Materials and Methods

### 2.1. Ethics Statement

Human specimens were obtained with written consent from patients under approval of the National Cancer Center Institutional Review Board (2002-69). Animal studies were carried out according to the Guideline for Animal Experiments in National Cancer Center, which meet the ethical standards required by the law and the guidelines about experimental animals in Japan, and approved by the Committee for Ethics in Animal Experimentation of the National Cancer Center.

### 2.2. Establishment of Primary Tumor Cell Culture

Clinical samples were obtained from the patients who underwent surgery for solid ovarian cancer neoplasms. Tumor tissues cut into small pieces (1 mm × 4 mm) were once frozen in TC protector cell freezing medium (DS Pharma Biomedical, Osaka, Japan) or directly digested in a Tumor Dissociation Kit, Human MACS (Miltenyi Biotech, Bergisch Gladbach, Germany). Dissociated tissue was subsequently filtered with a 70 μm pore cell strainer. From the primary single cell suspension, EpCAM positive cells were enriched by MACS microbeads (Miltenyi Biotech). The EpCAM positive cells were plated on 0.16 mg/mL Matrigel-coated plate (BD Biosciences, San Jose, CA, USA). Typically, the EpCAM-positive cells were seeded at a density of 1 × 10^4^~2 × 10^4^ per cm^2^ either in a 24-well, 12-well, 6-well plate or a 100 mm dish depending on the cell number. Though initially the EpCAM positive fraction was seeded, in most cases, flibroblast outgrowth was observed after the first passage. Fast (1–2 min) incubation with Accutase (Nacalai Tesque, Kyoto, Japan) was used to get rid of fibroblasts, but as it sometimes dissociated tumor cells at the same time the MACS EpCAM beads were repeatedly used then. After passage 2 when the fibroblasts were almost eliminated, the cells were seeded at the density of as low as 1 × 10^3^~2 × 10^3^ per cm^2^ depending on the doubling time of corresponding cell line. Medium was changed every three days. After establishment, sub-confluent cells were passaged at the ratio of 1:3 to 1:6 dilution every 3–10 days depending on the doubling time. Cells were always collected in DMEM:F12 supplemented with 5% FBS and 10 mM Y-27632. All cells were incubated in a humidified atmosphere containing 5% CO_2_ and 3% O_2_ at 37 °C.

### 2.3. Retrovirus and Lentivirus Infection

Preparation and infection of lentiviruses, CSII-CMV-tetOff and CSII-TRE-Tight-16E6E7-F2A-MYCT58A-F2A-HRASG12V, were described previously [12]. Retroviral vector, pRetroX-tetOnePuro-16E6E7-F2A-MYCT58A-F2A-HRASG12V, was constructed by recombining the 16E6E7-F2A-MYCT58A-F2A-HRASG12V segment into pDEST-RetroX-tetOnePuro by the LR reaction (Invitrogen, Carlsbad, CA, USA). Human RSPO1 and NOGGIN cDNAs were clones into PQCXIP (Clontech, Palo Alto, CA, USA) to generate PQCXIP-RSPO1 and PQCXIP–NOGGIN. Preparation and infection of retroviruses with 10A1 envelope were as described previously [12].

### 2.4. Cell Culture

Cells were cultivated in F-medium supplemented with Y-27632 [13] further supplemented with 10 μM SB202190, 10 mM nicotinamide, 500 nM A-83-01, 10 ng/mL bFGF and 5% (*v*/*v*) conditioned medium containing mouse Wnt3a, human R-Spondin 1(RSPO1) and human Noggin (medium A) or in Advanced DMEM/F12 (Life Technologies, Waltham, MA, USA) supplemented with 2 mM L-alanyl-L-glutamine solution (Nacalai Tesque), Wnt3a/RSPO1 conditioned medium (prepared in house) and Noggin (PeproTech, Rocky Hill, NJ, USA) or Wnt3a/RSPO1/Noggin conditioned medium (prepared in house), 10 mM, Nicotinamide (Wako, Kyoto, Japan), 10 µM p38 MAPK inhibitor SB-202190 (Selleck Chemicals, Houston, TX, USA), 10 µM Y-27632 (Selleck Chemicals-), 2% B27 and 100 ng/mL EGF (Sigma, St Louis, MO, USA). Both media were supplemented with 100 μg/mL Penicillin and 100 µg/mL Streptomycin (Nacalai Tesque). When the cells reached subconfluent, they were washed with PBS (Nacalai Tesque) and detached with Accutase (Nacalai Tesque). LWnt3A cells (ATCC CRL-2647) were transduced with PQCXIP-RSPO1 to produce Wnt3a/RSPO1 conditioned medium, and the resultant cells were further transduced with PQCXIP-NOG at MOI of 10 to produce Wnt3a/RSPO1/Noggin conditioned medium. Ovarian cancer cell line TOV112D was purchased from ATCC, and other ovarian cancer cell lines, RMG-1, ES2, A2780cis, OVK18, OVCAR3, SKOV3 and OV90 were obtained from Drs. Rumi Sasaki and Hidetaka Katabuchi (Kumamoto University) and cultivated in individual media recommended by ATCC in a humidified atmosphere containing 5% CO_2_ at 37 °C.

### 2.5. Cell Proliferation Assay

HOV5T (1 × 10^3^ cells/well), HOV26T (1 × 10^3^ cells/well), HOV28T (1 × 10^3^ cells/well), HOV29T (2 × 10^3^ cells/well) and HOV34T (1 × 10^3^ cells/well) cells were seeded on Matrigel-coated or uncoated (Matri-) 96 well plates with the complete medium B. Next day, the complete medium B was replaced with either medium A, F-medium, complete medium B or medium B without one component of the supplements, EGF, B27, Y27632, SB202190, nicotinamide, A83-01, Noggin conditioned medium, or Wnt3A and RSPO1 conditioned medium. Cells were fixed at four time points up to 13 days, and stained by acid red to measure the relative number of cells [14]. The reciprocal of the calculated doubling time in each condition was compared with that in complete medium B.

### 2.6. Western Analysis

Whole-cell protein extracts were used for western blotting as described previously [15,16] with primary antibodies listed in Appendix A. The LAS3000 CCD Imaging System (Fujifilm Co. Ltd., Tokyo, Japan) and Multi Gauge software (Fujifilm) were used for detection and quantification of proteins visualized by Lumi-light plus western blotting substrate (Roche, Basel, Switzerland).

### 2.7. Mice Xenograft Experiments

Established cell lines were suspended in PBS:matrigel (BD Bioscience) 1:1 solution and injected into the base of bilateral flank or peritoneum of female BALB/c nu/nu (Charles Rivers, Yokohama, Japan), NOD-SCID (Charles Rivers) or NOG (CLEA, Tokyo, Japan) mice. Mice were checked 2–3 times per week and sacrificed when the average volume reached 300 mm^3^. Those mice which did not develop tumors were observed for six months and then sacrificed. Mice were euthanized under isoflurane anesthesia and tumor tissues were excised and partly cut into small pieces and fixed with 10% formalin (Muto Pure Chemicals Ltd., Tokyo, Japan) followed by a preparation of paraffin-embedded sections and hematoxylin and eosin (HE) staining or immunohistochemistry.

### 2.8. Immunohistochemistry

Formalin-fixed paraffin-embedded (FFPE) tissue sections were de-waxed in xylene, rinsed well in ethanol, and then washed in tap water. Antigen retrieval condition, antibody and detection systems are listed in Appendix A.

## 3. Results

### 3.1. Establishment of Primary Tumor Cell Culture of Various Histotypes of Ovarian Cancer

Since there was not an efficient protocol for culturing all four main histotypes of ovarian cancer that do reproduce the original tumor histology properly, we aimed to generate a new method. Firstly, we cultured cells in the F-medium supplemented with Y-27632 [13] further supplemented with 10 μM SB202190, 10 mM nicotinamide, 500 nM A-83-01, 10 ng/mL bFGF and 5% (*v*/*v*) conditioned medium containing Wnt3A, RSPO1 and Noggin (medium A). We applied this culture condition to the first thirty ovary tumor specimens from HOV1T to HOV30T and only 5 cases out of 28 epithelial ovarian cancers (18%) were passaged more than five times (Table 1).

Most of the primary cultures underwent crisis after passage 1. Next, we applied medium B which consisted of Advanced DMEM/F12 (1:1 medium) supplemented with 100 ng/mL EGF, 10 μM Y27632, 5 μM SB202190, 10 mM nicotinamide, 500 nM A-83-01, 2% (*v*/*v*) B27, 1% (*v*/*v*) Glutamine and 5% (*v*/*v*) conditioned medium containing Wnt3A, RSPO1 and Noggin to the next nine ovarian tumor specimens from HOV31T to HOV39T including one germ cell tumor.

In this medium, we found the primary cells could proliferate better than those in medium A and eight cell lines out of eight eplithelial ovarian cancers (100%) were successfully established (Table 1), and the established cells formed compact colonies (Appendix A). Then we fed HOV28T and 29T cells which had been maintained in medium A until 2nd passage with medium B and established these cell lines. Then we applied this culture condition to the frozen tissues of four specimens and could establish three cell lines—HOV5T, 11T and 16T—though we failed to establish HOV30T. Although the primary culture was started with EpCAM-positive selected cells and the optimized culture condition had the capacity to repress growth of fibroblasts at a certain extent, fibroblast outgrowth was observed in most cultures. Further, EpCAM positive selection through microbeads or short accutase incubation was performed to get rid of fibroblasts. We refer to the ovarian cancer cell lines established by us as HOV, to distinguish them from those of other groups. After establishment, representative cell lines were cultured in different conditions including F-medium, medium A, B and medium B without each component and compared their growth (Table 2 and Appendix A). Matrigel-coated plates not only supported higher plating efficiency of the tested cell lines, but also enhanced the proliferation rate of HOV5T (Table 2 and Appendix A). EGF and B27 were almost essential for all types except for HGSC (HOV28T), and conditioned medium containing Wnt3a, RSPO1 and Noggin was specifically important for the proliferation of MC (HOV26T and HOV29T) cell lines. As nicotinamide was dispensable for two cell lines and rather strongly supressed growth of HOV5T and HOV29T, we currently use medium B without nicotinamide (Table 2 and Appendix A). Interestingly, HOV29T cells proliferated faster in F-medium than medium B, perhaps because F-medium did not contain nicotinamide. However, they stopped proliferation after a few passages under these culture conditions. F-medium accelerated the proliferation rate for a short term but seemed to induce differentiation as well and ultimately halted cell proliferation in long term cultures. HOV26T could hardly proliferate in medium F or A (Table 2). These results strongly confirmed the general superiority of medium B in combination with the usage of Matrigel-coated plates.

### 3.2. Recapitulation of Immunohistochemical Status of HOV Lines with the Original Tumor

According to the proposed algorithm [17] for the interpretation of immunohistochemical markers of ovarian cancer, WT1 positivity suggests serous carcinoma, although approximately 10% of HGSCs can be WT1-negative and aberrant TP53 and/or diffuse p16 (CDKN2A) expression are typical for HGSC, whereas a TP53 wild-type pattern and patchy p16 expression and WT1 positivity suggest a low-grade serous carcinoma. WT1 negative, PR positive suggests an EM carcinoma, although the sensitivity is only approximately 70%. CC carcinoma is PR negative, HNF1B positive and ARID1A (~50%) negative. EM and CC carcinomas usually show a wild-type pattern and absent/patchy p16 expression. In general, PAX8 expression is seen in three histotypes: HGSC, CC and EM. MC is likely to be negative for PAX8 as it is believed that MC originates from a tissue other than the Müllerian duct [8,18] CK7 and E-cadherin are commonly seen in epithelial ovarian tumors. Expression of ER is controversial, as there are reports showing that not all ovarian cancer specimens are ER positive, especially the EM type [19]. PR is rather nonspecifically expressed in histotypes other than the EM type [19].

Among the HOV lines that have been established, all except MC type were PAX8 positive (Figure 1). For HGSC, WT1 is seen in all three cases and modest accumulation of p53 protein, an indication of mutant p53, was observed. Among two HOV lines of EM type, both HOV5T and HOV19T were WT1 negative, matching the typical EM type expression pattern. Unusual expression of HNF1B in EM type was also observed in HOV5T. However, the original tumor of HOV5T was also positive for HNF1B (Figure 2B) suggesting the recapitulation of HOV lines with the original tumor. HNF1B was present and WT1 was absent in all seven CC type lines. ARID1A was absent in four out of seven CC lines and two out of two EM lines (Figure 1). All four MC-type HOV lines were negative for HNFB1 and WT1, and three out of four were negative for PAX8, while both HOV35 and its original tumor were positive for PAX8 (Figure 1 and Appendix A). Interestingly, one MC type line, HOV29T, showed ARID1A negative, which might indicate that the origin of this tumor is the same as that of CC and EM type tumors [20]. Expression of HNF1B in EM type clinical sample and cell line was also observed (Figure 1). E-cadherin was expressed except for one of the MC type lines which have sarcomatous component in original tumor (HOV26T, Figure 2A). HOV26T expressed high levels of N-cadherin as well as TP53, supporting the sarcomatous features of this cell line (Figure 1). Thus, the established cell lines maintained not only histotype-specific but also non-typical markers expressed in original tumors, suggesting that our cell lines could be used for ex-vivo ovarian cancer models.

Popular and common ovarian cancer cell lines highly cited in PubMed were examined for histotype-specific markers (Figure 1). Adenocarcinoma OV90 was negative for PAX8 and CK7, P53 overexpressed and HNF1B positive that does not suggest any histotype. OVK18, SKOV3 and ES2 did not express any type-specific markers. The expression manner of EM type A2780cis arose doubts about whether it is really an ovarian cancer cell line as none of the markers were expressed, except for self-keeping molecules. Marker expression pattern of TOV112D was nearly same as A2780cis. CC type RMG-1 expressed all the CC type specific markers. OVCAR3 showed comparatively better HGSC-like expression. Therefore, very few of the highly cited popular lines (two out of eight lines) showed ovarian cancer histotype-specific marker expression. Normal human tubal fimbria-derived cells immortalized with CDK4^R24C^, cyclin D1 and TERT [21] expressed WT1 and PAX8, though they did not express E-cadherin.

### 3.3. Pathological Fidelity of Xenograft Tissue from Established Lines

The gold standard of pathological fidelity of an established line is to recapitulate the original tumor tissue architecture in a xenograft tumor model. After confirming the in vitro histotype-specific fidelity, in vivo pathological fidelity was examined. Several cell lines representing all four major histotypes were injected into immunedeficient mice. Initially we subcutaneously injected two lines from CC type (HOV34T_ARID1A negative and HOV37T_ARID1A positive), two lines from EM (HOV5T_established in optimized condition and HOV19T_established in previous condition) type, one from HGSC (HOV28T) and one from MC (HOV20T) type as well as a highly cited ovarian cancer line, SKOV3, into nude mice. 

Also, to examine the tumorigenic and metastatic potential, we injected HGSC type HOV28T and EM type HOV5T cells as well as SKOV3 intraperitoneally into NOD-SCID mice. One (HOV34T-ARID1A negative) out of two CC type, one (HOV5T) out of two EM type and one HGSC type (HOV28T) subcutaneously injected cells formed tumors in nude mice. Intra-peritoneally injected HOV28T cells also generated tumor and disseminated to other organs (data not shown). MC type HOV20T failed to generate tumor subcutaneously. The conventional line SKOV3 also generated tumor by subcutaneous and intraperitoneal injection. Results of the xenograft experiments are summarized in Table 3.

Comparison of the histology between xenografts and original tumors were performed by FFPE, Hematoxylin-Eosin (HE) staining (Figure 2A). In the original tumor of HOV5T, typical features of EM type, including back to back glands organized around a central lumina surrounded by elongated malignant epithelial cells was clearly reproduced in the xenograft tissue. In the original tissue HOV28T, features of HGSC including a poorly differentiated papillary growth architecture and marked cellular atypia with dense nuclear chromatin staining was observed and also reproduced in the corresponding xenograft tissue. In the original tumor of HOV34T, hobnail cells with prominent nuclear atypia and abundant clear cytoplasm and stromal hyalinization, typical features of CC type, were also reproduced in the HOV34T xenograft tumor tissue. Mucinous adenocarcinoma with sarcomatous component HOV26T xenograft showed sarcomatous features consisting of pleomorphic spindle cells with hyperchromatic nuclei similar to sarcomatous lesion of the original tumor. Conventional cell line SKOV3 also generated tumors in nude mouse but the tumor tissue did not suggest any specific ovarian cancer histotype (Figure 2A).

### 3.4. MC Type Cancer Cells Could not Generate Any Tumor Even in NOG Mice

Five out of nine transplanted cell lines failed to generate tumors in immunodeficient mice. In particular, all the lines—HOV20T, HOV29T and HOV35T—derived from MC type tumors failed to generate any tumors, even when we transplanted them both subcutaneously and intraperitoneally into NOG mice [22] and to our best knowledge there is no report of a cell-derived xenograft (CDX) model of MC type ovarian cancer. Only HOV26T established from MC adenocarcinoma with sarcomatous component could generate tumors in a NOG mouse by subcutaneous injection. As the HOV26T xenograft tissue resembled the sarcomatous part of the original tissue faithfully which showed pleomorphic spindle cells with hyperchromatic nuclei (Figure 2A), it is likely that the line was established from the cells in the sarcomatous part of the original tumor.

### 3.5. Recapitulation of Histotype-Specific Markers In Vivo

To investigate histotype-specific marker expression more precisely in vivo, xenograft tumors as well as the original tumor tissue were examined by immunohistochemical staining. Mouse xenograft tumor tissue completely recapitulated the original tissue in terms of expression of the histotype-specific as well as non-specific markers (Figure 2B). For example, the HGSC type HOV28T was WT1 negative unlike typical HGSC, but the original tumor tissue itself was WT1 negative, indicating the xenograft tissue well mirrored the original tissue. Similarly, both the xenograft and original tumor tissue of HOV5T expressed HNF1B though typical EM type tumors don’t. Apart from this almost all the histotype-specific marker expression was confirmed in both mouse xenograft and human tumors (Figure 2B and Appendix A). However, the conventional line SKOV3 reported to be established from acites of ovarian adenocarcinoma [23] generated xenograft tumors whose tissue architecture did not resemble any typical histotype though cellular atypia with clear cytoplasm might imply CC type. Immunohistochemically, they showed WT1 negative and HNF1B positive implying CC-type but p53 negative implying rather HGSC. Therefore, the SKOV3 xenograft does not provide a model for any typical histotype of ovarian cancer, which is consistent with its mRNA expression profile [3].

### 3.6. Establishment of Xenograft Models for MC and Endometrioid Ovarian Cancers by Conditionally Expressing Additional Set of Oncogenes

To our best knowledge there is no report about CDX from MC epithelial type ovarian cancer. In our case, none of the three epithelial type lines established was able to generate tumors within 6 months after injection, even in NOG mice. Also, among two EM type lines, HOV19T failed to generate tumor in nude mouse. Based on our previous report that transduction of human papillomavirus type 16 (HPV 16) E6 and E7 genes (E6E7), MYC and oncogenic HRAS (HRASG12V) (EMR) was sufficient for tumorigenic transformation of normal human cells, such as cervical keratinocytes, tongue keratinocytes, bronchial epithelial cells and pancreatic epithelial cells [12,24,25,26]. We hypothesize that aberrant and conditional expression of the set of oncogenes in poorly tumorigenic cancer cell lines in mice should enhance xenograft tumor formation and shut down of EMR genes in vivo may lead to reconstruction of tissue architecture corresponding to the original tumor tissues.

A polycistronic lentivirus vector, CSII-TRE-Tight-16E6E7-F2A-MYCT58A-F2A-HRASG12V, designed to express E6E7, MYC and HRASG12V (EMR) as well as CSII-CMV-tetOff vector was transduced into typical MC type lines, HOV20T and HOV29T, and cultivated in the absence of doxycycline (DOX) so that the EMR genes are expressed (tetOff system). An endometrioid type line, HOV19T, was transduced with a tet-inducible polycistronic retrovirus vector, pRetroX-tetOnePuro-16E6E7-F2A-MYCT58A-F2A-HRASG12V, selected in the presence of 1 μg/mL puromycin and cultivated in the presence of 1 μg/mL of DOX (tetOn system). DOX-dependent expression of the oncogenes was confirmed by western blotting (Figure 3A). As the doubling time of HOV29T was about 7–8 days in medium B (Appendix A), cell growth was greatly enhanced by the expression of transgenes in HOV29T-EMR cells (Figure 3B). When these cells were subcutaneously injected into four nude mice at four sites per mouse in a condition where the EMR genes were expressed, they formed tumors as expected. When the average tumor size exceeded 300 mm^3^, DOX was administered (HOV20T and 29T) or withdrawn (HOV19T) in drinking water of mice, which resulted in regression of tumor growth (Figure 3C). Mice transplanted with HOV29T-EMR cells, were sacrificed either on day 48 without DOX administration or on day 4, 12 and 22 after DOX administration. Mouse tumor tissue without DOX administration exhibited poorly differentiated cancer morphology with least MC histology. 

The xenograft tissue 4 days post-DOX administration showed intermediate histology between poorly and moderately differentiated cancer. However, the tissue 12 days post-DOX administration showed typical MC type histology with single-layered epithelium which completely recapitulated the original tumor tissue (Figure 3D). Essentially the same results were obtained with HOV20T and HOV19T, both of which reproduced tumor histology faithfully recapitulated the original tumor tissue after shutting-off the EMR genes (Appendix A). Since the pathogenesis and biologic behavior of MC epithelial type differ substantially from non-MC histology of ovarian cancer, this CDX model will provide useful information to the scientific community for understanding the MC type ovarian cancer biology. This strategy could be applicable to not only cancer cell lines but also cell lines derived from precancerous or even normal tissues, which are difficult to generate xenograft tumors. Indeed, HOV29T is derived from MC borderline tumor.

## 4. Discussion

In this report, we have described a robust method to establish ovarian cancer cell lines from primary sites with around 95% efficiency. HOV11T was established from an ovarian tumor which was diagnosed as adenocarcinoma NOS, it could be classified as HGSC since HOV11T cells showed accumulation of TP53 and p16 and WT positivity (Figure 1). Thus we have established 17 ovarian tumor cell lines belonging to all four major histotypes including seven CC type, two EM type, three HGSC, one possible HGSC and three MC type lines, as well as a sarcomatous line derived from a MC tumor with sarcomatous components. Specific markers for four main ovarian carcinomas were examined in these lines, and compared with those with original tumors as well as xenograft tumors. Through this ex-vivo culture system, the in vivo status phenotype could be maintained in vitro. However, not all cell lines could form xenograft tumors. In particular, MC type cell lines failed to generate tumors even in NOG mice. To our knowledge, no xenografts from cell lines derived from MC borderline tumors or MC adenocarcinomas have been reported and neither could we by our own hand. To establish CDX models from these cell lines, we developed a novel method which conditionally enhanced the tumorigenicity of those poorly tumorigenic cell lines, including a MC adenocarcinoma, a MC borderline tumor and an EM tumor cell line, by conditionally expressing a set of additional oncogenes in the cells. Cells expressing a set of oncogenes formed tumors showing poorly differentiated histology different from the original tumors. However, after silencing the oncogenes regulated by doxycycline administration to the mice, the tumor size decreased and the remaining tumor tissues restored histology of the corresponding original tumor faithfully (Figure 3D and Appendix A). Mucin accumulation is a critical feature for diagnosis of MC tumors. A previous study indicated that ovarian mucinous tumors show similar mucin immune-profile being strongly and uniformly positive for MUC5AC irrespective of whether they are benign, borderline or malignant, but only focally positive for MUC1, MUC2, and MUC6, which is different from pancreatic, biliary, esophageal, gastric, and colorectal/appendiceal carcinomas [27,28]. Thus the observed mucin accumulation in xenograft tumors of HOV20 and HOV29T is likely to be MUC5AC. Though the cells-of-origin of MC tumors is still unknown [8], our method will be useful to clarify this important question. Combination of HPV16 E6, E7, MYC and activated HRAS were used as additional oncogenes as we previously had shown that these oncogenes were sufficient to convert several normal human epithelial cells into tumorigenic cells [12].Thus it is likely this combination can convert any non-tumorigenic ovarian cancer cell lines in mice into tumorigenic.

The success rate of primary cultures depends on the culture conditions. We thought that coating with extracellular matrix and supplementation with appropriate growth factors and inhibitors that can mimic the in vivo environment and reduce the culture stress were important to retain original features of cells through an ex-vivo culture system. So far there are several reports about establishment of ovarian cancer cells from clinical samples and their success rate is not so high. So far the best report with 95% efficiency established for only nearly 50% of lines from primary tissue and the rest were from xenograft or ascites [10], and also histotype-specific marker expression comparing with original sample could not be observed whereas we achieved 100% (8/8) efficiency from fresh primary tumors and 80% (4/5) from frozen specimens whose fresh specimens failed to establish cell lines with other methods. The only case, HOV30T, we failed to establish was an adenocarcinoma after chemotherapy and the amount of starting material available was very small. Three cell lines established with another culture conditions could be also maintained under the current culture conditions. As we enriched EpCAM-positive cells to establish new HOV cell lines, we cannot formally exclude the possibility that we excluded EpCAM-negative cancer cells and included normal epithelial cells. However, every one of our cohorts showed a well-differentiated histology and we obtained similar histology in xenografts through our cell culture method. Moreover, we could also establish a HOV26T sarcomatous cell line, which was probably EpCAM-nagative, from the sarcomatous component in the original tumor. Thus, these results lead us to conclude that our cell culture system is a superior method which is useful in terms of faithful recapitulation of the original molecular and histological features.

Nowadays it is believed that effective therapy depends on the identification of functionally active cells associated with drug-resistance and self-renewal properties which cancer or tumor initiating cells (CSC) generally possess. So far there are several reports [29,30] that have identified CSC markers of ovarian cancer but most of those were identified in clinical and conventional cell lines that were established under physiologically irrelevant conditions. As the HOV cell lines established by our protocol contain CSCs to generate xenograft tumors recapitulating the original tumor, they will help to understand ovarian CSC biology by checking the expression of CSC markers as well as actual tumorigenicity.

PDX models are well known to preserve key biological properties of primary tumors but at the same time it is a time consuming and expensive methid. In additon, the success rate is not very high [31]. Also, recently, there is a report that most commonly used cancer cell lines have no correlation with the original histotype [4]. As the original marker expression regardless of histotype specificity was mirrored in xenografts, our robust and inexpensive culture system that preserves the molecular markers faithfully and the CDX model will be able to replace the necessity of PDX models and commercially available ovarian cancer cell lines.

Understanding ovarian cancer biology is more difficult than others, as each histotype has been thought to be derived from different histology and have their own tumorigenic pathways. We could establish cell lines as well as CDX models from all four major histotypes. An authentic cellular model is a first and foremost need for preclinical validation assay of novel targeting agents. These highly authentic seventeen lines will make preclinical assays easier for the scientific community.

## Figures and Tables

**Figure 1 cells-08-00644-f001:**
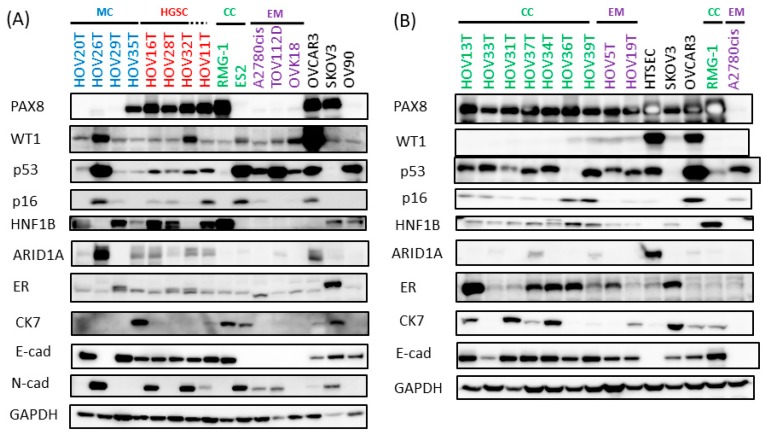
Expression of ovarian cancer several histotype-specific markers in established HOV cell lines and commercially available ovarian cancer cell lines. (**A**) HOV cell lines from mucinous (MC) carcinoma and high-grade serous adenocarcinoma (HGSC) (**B**) HOV cell lines from clear cell (CC) and endometrioid (EM) carcinomas Expression of PAX8, p16, CK7, E-cadherin, HNF1B, ARID1A, WT1, p53 and ER as well as GAPDH as a loading control were detected by Western blotting. Expression of N-cadherin was also detected to examine the sarcomatous feature of HOV26T in (A). RMG-I and ES2 are CC carcinoma cell lines; A2780cis, TOV112D and OVK18 are EM carcinoma cell lines; OV90 is a papillary serous carcinoma cell line; SKOV3 and OVCAR3 are unclassified adenocarcinoma cell lines.

**Figure 2 cells-08-00644-f002:**
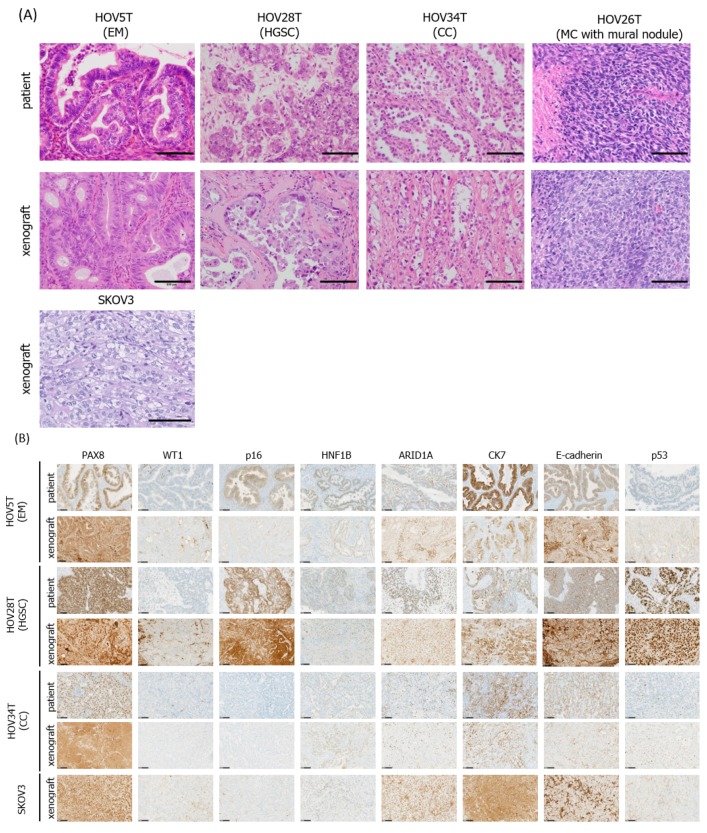
Recapitulation of original tumor tissue architecture and histotype-specific marker expression in xenografts of HOV cell lines. HOV lines and highly cited ovarian cancer cell line SKOV3 were subcutaneously injected to immunocompromised mice. (**A**) FFPE tissue sections from the xenograft tumors of HOV5T, HOV28T, HOV34T, and SKOV3 and original tumors corresponding to HOV cell lines were stained with hematoxylin and eosin. Representative microscopic images are shown. Scale bar 100 μm. (**B**) FFPE sections of patient tumor tissue, HOV xenograft and SKOV3 xenograft were stained with several ovarian cancer histotype-specific markers: PAX8, Wilm’s tumor 1(WT1), p53, p16, HNF1B, ARID1A (these are nuclear staining) and Cytokeratin 7 (CK7) (cytoplasmic, membranous staining), E-cadherin (inter-cellular junction). Note that the occasional cytoplasmic signals in nuclear staining markers and nuclear signals in CK7 and E-cadherin staining are likely to be non-specific. Scale bar 100 μm.

**Figure 3 cells-08-00644-f003:**
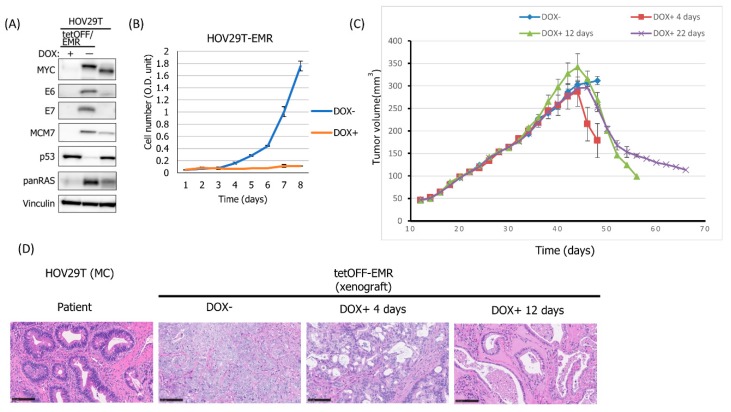
Generation of tumors and recapitulation of original tumor tissues by temporal expression of additional oncogenes in MC type HOV line. (**A**) Expression of the transgenes and their surrogate markers as well as vinculin as a loading control were detected by western blotting. (**B**) Cell growth curves of HOV29T/tetOFF/EMR cells with or without DOX. Each point is the mean of the triplicates ± SEM. (**C**) In vivo tumor generation capability of HOV29T/tetOFF/EMR cells. Cells were subcutaneously injected into nude mice. At day 44 when the average volume of tumors injected at 4 sites per mouse reached 300mm3, three mice were fed with drinking water supplemented with DOX (1 mg/mL) and sacrificed at 4 days (red), 12 days (green) and 22 days (purple) post-administration. Each point is the mean of the tumor volume ± SEM (n = 4). (**D**) Histopathology of HOV29T/tetOFF/EMR xenograft and original tumor of HOV29T, H&E. Scale bar 100 μm

**Table 1 cells-08-00644-t001:** Summary of established cell lines.

**Code No.**	**Age**	**Type of cancer**	**Medium A**	**A ≥ B**	**Medium B**
**HOV3T**	**47**	**Adenocarcinoma, residual** **MC cystadenoma**	**Failed**		
**HOV4T**	**42**	**Adenocarcinoma, residual** **MC cystadenoma**	**Failed**		
**HOV5T**	**43**	**EM**	**Failed**		**✓**
**HOV6T**	**59**	**CC**	**Failed**		
**HOV7T**	**59**	**HGSC**	**Failed**		
**HOV8T**	**61**	**HGSC**	**Failed**		
**HOV9T**	**52**	**HGSC**	**Failed**		
**HOV10T**	**40**	**MC**	**✓**		
**HOV11T**	**46**	**Adenocarcinoma**	**Failed**		**✓**
**HOV12T**	**49**	**MC borderline**	**Failed**		
**HOV13T**	**47**	**CC**	**Failed**		**✓**
**HOV14T**	**49**	**HGSC**	**Failed**		
**HOV15T**	**49**	**MC cystadenoma**	**Failed**		
**HOV16T**	**50**	**HGSC**	**Failed**		**✓**
**HOV17T**	**67**	**EM**	**Failed**		
**HOV18T**	**67**	**MC cystadenoma, borderline**	**Failed**		
**HOV19T**	**35**	**EM**	**✓**		
**HOV20T**	**49**	**MC**	**✓**		
**HOV21T**	**50**	**HGSC**	**Failed**		
**HOV22T**	**45**	**HGSC**	**Failed**		
**HOV23T**	**62**	**MC borderline**	**Failed**		
**HOV24T**	**56**	**Adenocarcinoma (post-chemo)**	**Failed**		
**HOV25T**	**41**	**EM borderline**	**Failed**		
**HOV26T**	**74**	**MC with sarcomatous component**	**Aborted**	**✓**	
**HOV27T**	**74**	**HGSC**	**Failed**		
**HOV28T**	**28**	**HGSC**	**Aborted**	**✓**	
**HOV29T**	**40**	**MC borderline**	**Aborted**	**✓**	
**HOV30T**	**62**	**Serous (post-chemo)**	**Failed**	**Failed**	**Failed**
**HOV31T**	**71**	**CC**	**Aborted**		**✓**
**HOV32T**	**67**	**HGSC**	**Aborted**		**✓**
**HOV33T**	**36**	**CC**	**Aborted**		**✓**
**HOV34T**	**37**	**CC**	**Aborted**		**✓**
**HOV35T**	**68**	**MC borderline with micro invasion**	**Aborted**		**✓**
**HOV36T**	**66**	**CC**	**Aborted**		**✓**
**HOV37T**	**59**	**CC**	**Aborted**		**✓**
**HOV39T**	**68**	**CC**	**Aborted**		**✓**

Notes: HOV1T (metastatic tumor of colorectal cancer), HOV2T (adenofibroma) and HOV38T (mixed germ tumor) were omitted from this table since they were not epithelial ovarian cancers. All tissue sources were resected primary solid tumors. Aborted, aborted when we noticed medium B supported proliferation of the cells better than medium A; A ≥ B, primarily cultivated in medium A and then in medium B no later than at passage 3; **✓**, established.

**Table 2 cells-08-00644-t002:** Effect of individual components of medium B, different media or Matrigel-coating on proliferation of HOV lines.

	**Matri **	**EGF **	**B27 **	**Y **	**SB **	**NA **	**A83 **	**Nog **	**WR **	**A **	**F **
**HOV5T (EM)**											
**HOV26T(MC)**											
**HOV29T(MC)**											
**HOV28T(HGSC)**											
**HOV34T(CC)**											

					**3−**	**2−**	**1−**	**0**	**1+**	**2+**	**3+**

Notes: Cells were cultivated in complete medium B, medium B without each component, medium A or F-medium and the cell proliferation rate (the reciprocal of doubling time) was calculated from Appendix A and compared with complete B medium on the Matrigel-coated plate: 0: 80–120% (white), 1+: 60–80% (light orange), 2+: 40–60% (orange), 3+: <40% (red), 1-: 120–150% (light blue), 2-: 150–200% (blue), 3-: >200% (dark blue) increase for each component or decrease for medium A and F-medium. Matri: Matrigel-coating, Y: Y27632, SB: SB202190, NA: nicotinamide, A83: A-83-01, Nog: Noggin conditioned medium, WR: Wnt3A and RSPO1 conditioned medium, A: medium A, F: F-medium.

**Table 3 cells-08-00644-t003:** Summary of xenograft transplantation of HOV lines and SKOV3.

**Code No.**	**Type of Cancer**	**Nude** **s.c.**	**SCID** **i.p.**	**NOG** **s.c.**	**NOG** **i.p.**	**Nude** **s.c. (EMR)**
**HOV5T**	**EM**	**6/6 (9)**	**0/1**			
**HOV19T**	**EM**	**0/4**				**4/4(9)** **4/4(11)** **4/4(13)**
**HOV34T**	**CC**	**4/4 (5)**				
**HOV37T**	**CC**	**0/4** **0/4**				
**HOV20T**	**MC**	**0/4**		**0/4**	**0/1**	**4/4(18)** **4/4(18)** **4/4(21)** **4/4(21)**
**HOV26T**	**MC with sarcomatous component**			**4/4 (14)**	**0/1**	
**HOV29T**	**MC borderline**			**0/4**	**0/1**	**4/4 (7)** **4/4(7)** **4/4(8)** **4/4(10)**
**HOV35T**	**MC borderline**			**0/4**	**0/1**	
**HOV28T**	**HGSC**	**4/4 (7)**	**1/1 (21)**			
**SKOV3**		**4/4 (7)**	**1/1 (6)**			

Notes: Incidence of tumor formation within 6 months of observation period was scored. Fractional expression means (number of tumors generated)/(number of sites injected) in each mouse. Number in parentheses indicates observation period (weeks) when mice were sacrificed, and no parenthesis indicates mice were sacrificed after 6 months. s.c., subcutaneous injection; i.p., intraperitoneal injection; EM, endometrioid carcinoma; CC, clear cell carcinoma; MC, mucinous carcinoma; HGSC, high-grade serous carcinoma; EMR, with E6E7, MYC and RAS oncogenes (see Result for details).

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
