# Peer review of "An Ex-Vivo Culture System of Ovarian Cancer Faithfully Recapitulating the Pathological Features of Primary Tumors"

_cells, 2019, doi:10.3390/cells8070644_

Round 1
Reviewer 1 Report
The manuscript by Ghani et al. titled "An ex-vivo culture system of ovarian cancer faithfully recapitulating the pathological features of primary tumors" is an interesting basic sciences study to develop new ovarian cancer cell lines. The methods are intriguing and although the study could help future research on the field it would benefit from addressing the following concerns:
1) More detailed culture conditions (cell densities, media change schedules, cultures times, etc) must be included to understand what was done and be able to repeat the results.
2) Another method to estimate the cell expansion under different culture conditions should be included.
3) Numbers of injected mice should be included.
4) Table 1 is somewhat confusing in regards to what was actually done to the cells.
5) Table 2 is not an informative way to present the data. Column graphs of the actual data should be presented including the error bars and significance levels.
6) For the mice xenograft data more information needs to be included. How many mice were used? How many mice were positive for tumor formation? How many were negative?
7) In Figure 3C, statistical significance must be calculated and shown to draw the conclusions written in the results.
Author Response
Response to the Reviewer 1’s comments
Thank you for your positive and helpful comments on our manuscript.
Point 1: More detailed culture conditions (cell densities, media change schedules, cultures times, etc) must be included to understand what was done and be able to repeat the results.
Response 1: Thank you for the comments. Following the suggestion, we added detailed culture conditions in page 3 from line 100.to 107
Point 2: Another method to estimate the cell expansion under different culture conditions should be included.
Response 2: Thank you for your suggestion. We have described the method in the figure legend, but we added the method for the cell expansion under different culture conditions in page 4 line 135 to 138 as follows.
Cells were seeded on matrigel-coated or uncoated 96 well plates with the complete medium. Next day, the cells were fed with different media. Cells were fixed at four time points up to 13 days, and stained by acid red to measure the relative number of cells [14]. The reciprocal of the calculated doubling time in each condition was compared with that in complete medium B.
Point 3: Numbers of injected mice should be included.
Response 3: Thank you for your suggestion. As we injected cells in multiple sites of the same mouse, numbers of sites injected and number of tumors generated were added in the Table 3.
Point 4: Table 1 is somewhat confusing in regards to what was actually done to the cells.
Response 4: Thank you for the comment. We modified Table 1 and its footnote to demonstrate the results more clearly.
Point 5: Table 2 is not an informative way to present the data. Column graphs of the actual data should be presented including the error bars and significance levels.
Response 5: Thank you for the comments. Actual data are shown in Extended Figure A2. However, as the cell proliferation graphs are too complex to interpret, we decided to add the table in Figure A2. To make the table more comprehensive, we changed it as a heatmap as suggested by reviewer 2.
Point 6: For the mice xenograft data more information needs to be included. How many mice were used? How many mice were positive for tumor formation? How many were negative?
Response 6:Thank you for your suggestion. As we injected cells at multiple sites of the same mouse, number of sites injected and number of tumors generated were added in the Table 3.
Point 7: In Figure 3C, statistical significance must be calculated and shown to draw the conclusions written in the results.
Response 7: In this experiments, histological changes after doxycycline administration are critical and the size of the tumors are not. It is crystal clear that the tumor growth was inhibited by the doxycycline administration and we believe that calculation of statistical significance of tumor growth has no merit to draw our conclusions.
Reviewer 2 Report
The manuscript entitled “An ex-vivo culture system of ovarian cancer faithfully recapitulating
the pathological features of primary tumors “ by Ghani et al. presents a novel culture method which can support proliferation of primary epithelial ovarian cancer cells for cell line development. The authors performed significant amount of works and characterization. However, some more explanation and discussion regarding the results should be improved. It is recommended that this paper should be significantly revised before being accepted. Detailed comments are listed below:
1. The authors used MACS microbeads to enrich EpCAM positive cells. However, it is possible that the proposed method will exclude EMT-like EpCAM negative cells, which are critical for cancer metastasis. Also, is it possible that normal epithelial maybe collected in the cell culture? It is recommended that the authors can comment on those issue with supporting data or previous literature.
2. Following previous question, the authors used short-term incubation with Accutase to remove fibroblasts. It is recommended that the authors can provide data to show the enrichment efficacy and impurity of this method. The experiment may be demonstrated by conventional cancer cell line versus fibroblast cell line.
3. In discussion, the authors discussed the potential impact of this work regarding CSC. However, CSC markers were not measured and compared in this study. The argument does not seem to be supported by data.
4. “Milteny” Biotech seems to be a typo.
5. The meaning of “Abandoned” in Table 1 was not explained. It is recommended that the authors can specifically comment on why those samples were abandoned.
6. The data presentation of Table 2 is not straightforward. It is recommended that the authors may present that with color heatmap or other manners, so it maybe easier for audience to follow.
Author Response
Response to the Reviewer 2’s comments
The manuscript entitled “An ex-vivo culture system of ovarian cancer faithfully recapitulating
the pathological features of primary tumors “ by Ghani et al. presents a novel culture method which can support proliferation of primary epithelial ovarian cancer cells for cell line development. The authors performed significant amount of works and characterization. However, some more explanation and discussion regarding the results should be improved. It is recommended that this paper should be significantly revised before being accepted. Detailed comments are listed below:
Thank you for your positive and helpful comments to improve our manuscript.
Point 1: The authors used MACS microbeads to enrich EpCAM positive cells. However, it is possible that the proposed method will exclude EMT-like EpCAM negative cells, which are critical for cancer metastasis. Also, is it possible that normal epithelial maybe collected in the cell culture? It is recommended that the authors can comment on those issue with supporting data or previous literature.
Response 1: We agree to the reviewer’s comment that it is possible that the proposed method will exclude EMT-like EpCAM negative cells and we cannot formally exclude the possibility. However, our every cohort showed a well-differentiated histology, and we gained the similar histology through our cell culture method. Moreover, sarcomatous component which is seen in HOV26T are also confirmed to exist in the tumor of its origin. We think that these results lead to the conclusion that our cell culture system is useful in terms of recapitulation of its original histology faithfully. To emphasize this point, we changed the term “mural nodule “ to “sarcomatous component” in this manuscript. Sometimes heterogeneity can be seen within one tumor lesion. At present, however, even in our improved culture condition, contamination of mesenchymal cells are often detrimental for the growth of tumor cells. We think it is a next challenge to examine whether such heterogeneity could be maintained in our culture system and to find better methods to enrich or selectively enhance tumor cell growth.
Point 2: Following previous question, the authors used short-term incubation with Accutase to remove fibroblasts. It is recommended that the authors can provide data to show the enrichment efficacy and impurity of this method. The experiment may be demonstrated by conventional cancer cell line versus fibroblast cell line
Response 2: To be honest, this method is easy but not so effective or even not applicable to some cases, as it depends on differential detachment from the culture dish between tumor cells and fibroblasts and some tumor cells are as sensitive as fibroblasts. In such cases, we used the EpCAM MACS microbeads. To clarify the fact, we added a sentence in page 3 line 103 as follows.
Fast incubation with Accutase sometimes dissociated tumor cells at the same time and in such case the MACS EpCAM beads were used.
Point 3: In discussion, the authors discussed the potential impact of this work regarding CSC. However, CSC markers were not measured and compared in this study. The argument does not seem to be supported by data.
Response 3: CSC markers are surrogate markers to indicate stemness of CSCs. Since non-CSCs are unable to make tumors, tumorigenicity of the cells itself is direct evidence of inclusion of CSCs. However, as we have not done transplantation of smaller numbers of cells, we have no data about frequency of CSCs in our cell lines. Thus we changed statements about the potential impact of this work regarding CSC as follows.
As the HOV cell lines established by our protocol contain CSCs to generate xenograft tumors recapitulating histological features of original tumor, they will help to understand ovarian CSC biology by checking the expression of CSC markers as well as actual tumorigenicity.
Point 4: “Milteny” Biotech seems to be a typo.
Response 4: Thank you for the comment on our careless mistake. We corrected the typo.
Point 5: The meaning of “Abandoned” in Table 1 was not explained. It is recommended that the authors can specifically comment on why those samples were abandoned.
Response 5: Thank you for the comment. We aborted the cells cultivated in medium A when we noticed cell growth in medium B was much better than that in medium A. As “Aborted” instead of “Abandoned” might be a better word for it, we added the meaning of “Aborted” in the footnote as follows. Aborted, aborted when we noticed medium B supported proliferation of the cells better than medium A.
Point 6: The data presentation of Table 2 is not straightforward. It is recommended that the authors may present that with color heatmap or other manners, so it maybe easier for audience to follow.
Response 6: Thank you for the comment. Actual data are shown in Extended Figure A2. However, as the cell proliferation graphs are too complex to interpret, we decided to make this table. Following the comment and to make the results more comprehensive, we changed it as a heatmap table.
Reviewer 3 Report
The paper by Ghani et al. described new culture systems to obtain ovarian cancer cell lines from fresh tumor samples. Theapper address an important point to obtain new preclinical in vitro and in vivo models to study ovarian carcinoma. There are however few points that need to be fulfil.
Table 1: what is the meaning of abandoned’
It is not clear for how many passages were all the cell lines cultured. Five? Were was any difference in cell growth among the different passages?
At which passage were the westerns reported in Figure 1 made? In addition the figure should report the western data on the original tumors as not in all the cases the data are available. In the opinion of this referee the recapitulation of immunohistochemical status of the new HOV cell lines has to be shown at first with the primary tumors and then contextualized in general in the immunohistochemical markers of ovarian carcinoma
As regards the in vivo experiemnts it is not clear how many mice were transplanted and in how many the tumors appeared. In fact the % of tumor take is another important point. Were again tumorigenic the xenogarfts when transplanted in other recipient mice?
The experiments on transduction of the polycistronic vector to render tumorigenic need to be explained.
Cells tranduced in vitro seem to growth very much when given DOX, while the cells not given DOX are not growing . This is strange as it was previously stated that cells lines were obtained and this implied that cells were growing. In any case the transduction of the vector could alter the cells and the xenografts derived from them In addition the only histological examination (hematossilin and eosin) is too scanty to say that the growing tumor recapitulated the cells from which they derived. The positivity of a panel of mucionous proteins should be shown as compared to the original primary tumors
Author Response
Response to the Reviewer 3’s comments
Thank you for your positive and helpful comments to improve our manuscript.
Point 1: Table 1: what is the meaning of abandoned’
Response 1: Thank you for the comment. We aborted the cells cultivated in medium A when we noticed cell growth in medium B was much better than that in medium A. As “Aborted” instead of “Abandoned” might be a better word for it, we added the meaning of “Aborted” in the footnote as follows.
Aborted, aborted when we noticed medium B supported proliferation of the cells better than medium A.
Point 2: It is not clear for how many passages were all the cell lines cultured. Five? Were was any difference in cell growth among the different passages?
Response 2:. We cultivated the cells until the cells appeared monotonous without fibroblasts and propagated in multiple 10 cm dishes for freeze stocks as well as cell lysates. Sometimes we can obtain such population just after one passage though typically it took 5~6 passages. We counted cell numbers in each passage, but the numbers included fibroblasts in earlier passage, it is difficult to compare cell growth in different passages.
Point 3: At which passage were the westerns reported in Figure 1 made?
Response 3:. As described in the above, cell lysates harvested at different passages were used for western blitting. Most cell lysates were harvested at passage 4 to 6 except for HOV32T (HGSC) was harvested at passage 1 and HOV26T was harvested at passage 12.
Point 4: In addition the figure should report the western data on the original tumors as not in all the cases the data are available.
Response 4:. As the original tumors contain not only tumor cell but also varieties of different cell types including blood cells as well as extra cellular matrix, we think it is more suitable for IHC analyses but not for Western blotting.
Point 5: In the opinion of this referee the recapitulation of immunohistochemical status of the new HOV cell lines has to be shown at first with the primary tumors and then contextualized in general in the immunohistochemical markers of ovarian carcinoma
Response 5: Thank you for the suggestion about the data presentation. It may be a good idea to demonstrate our main conclusion about the CDX model. However, we applied only several cell lines to the CDX model and as other reviewers do not suggest such change and we would like to state the results as they are.
Point 6: As regards the in vivo experiemnts it is not clear how many mice were transplanted and in how many the tumors appeared. In fact the % of tumor take is another important point. Were again tumorigenic the xenogarfts when transplanted in other recipient mice?
Response 6: Thank you for your suggestion and we agree to the comments. As we subcutaneously transplanted cells in multiple sites per mouse, numbers of sites injected and number of tumors generated were added in the Table 3.
Point 7: The experiments on transduction of the polycistronic vector to render tumorigenic need to be explained.
Response 7: Thank you for the comment. Though we provided essential procedures in Methods section, we added more detailed experimental procedure in page 13 line 355 to 366 as follows.
A polycistronic lentivirus vector, CSII-TRE-Tight-16E6E7-F2A-MYCT58A-F2A-HRASG12V, designed to express E6E7, MYC and HRASG12V (EMR) as well as CSII-CMV-tetOff vector was transduced into typical MC type lines, HOV20T and HOV29T, and cultivated in the absence of doxycycline (DOX) so that the EMR genes are expressed (tetOff system). An endometrioid type line, HOV19T, was transduced with a tet-inducible polycistronic retrovirus vector, pRetroX-tetOnePuro-16E6E7-F2A-MYCT58A-F2A-HRASG12V, selected in the presence of 1 g/ml puromycin and cultivated in the presence of 1 g/ml of DOX (tetOn system). DOX-dependent expression of the oncogenes was confirmed by western blotting (Figure 3A). Cell growth was also controlled by transgenes in HOV29T-EMR cells (Figure 3B). Cell growth in the presence of DOX was slower than parental HOV29T cells possibly due to their addiction to the EMR gene. When these cells were subcutaneously injected into four nude mice at four sites per mouse in a condition where the EMR genes were expressed, they formed tumors as expected.
Point 8: Cells transduced in vitro seem to growth very much when given DOX, while the cells not given DOX are not growing. This is strange as it was previously stated that cells lines were obtained and this implied that cells were growing. In any case the transduction of the vector could alter the cells and the xenografts derived from them. In addition the only histological examination (hematossilin and eosin) is too scanty to say that the growing tumor recapitulated the cells from which they derived. The positivity of a panel of mucionous proteins should be shown as compared to the original primary tumors.
Response 8: Thank you for the thoughtful comments. We agree to the reviewer’s comment that this looks strange as it was previously stated that cells lines were obtained and this implied that cells were growing. However, as the proliferate rate of HOV29T was exceptionally low and the actual doubling time in medium B was about 9 days, the cell growth in the presence of DOX was almost unchanged whereas the doubling time of the EMR-expressing HOV29 was about 28 hours indicating the transgene expression greatly enhanced its proliferation rate, which is indeed another merit of conditional expression of transgenes. To clarify this point, we modified a sentence in page 13 line 362 to 363 as follows.
We also agree to the reviewer’s comment that the transduction of the vector could alter the cells and the xenografts derived from them, and this could be a limitation of the current method. Finally, we agree to the reviewer’s critical comment that the only histological examination (hematoxilin and eosin) is too scanty to say that the growing tumor recapitulated the cells from which they derived and IHC analyses and other biological examination will be required to see whether the cell line recapitulate other features of original tumor cells which is an important scope of future study. However, we think this experiment showed a very strong impact that this DOX system restored not only the ability of mucin production but also the tumor histology itself in which all factors such as cell adhesion and/or cell structure were reflected. In that respect, we think that the most convincing and most impressive figure is to show the histology which is a culmination of all molecular pathological pathway and/or phenomena. Furthermore, to our knowledge, as xenograft models for mucinous ovarian carcinoma have not been recapitulated even at this level our method has still some advantage to make xenograft models from mucinous-type ovarian tumors.
Round 2
Reviewer 1 Report
Thank you for responding to my suggestions and making the appropriate changes to the manuscript. The manuscript is much better now.
Author Response
Point 1. Thank you for responding to my suggestions and making the appropriate changes to the manuscript. The manuscript is much better now.
Response 1. We appreciate very much for your editorial work and positive comments.
Reviewer 2 Report
The authors gave good explanation to the reviewers but did not include much explanation in the manuscript. It is recommended that the authors include more discussion in the manuscript to improve the manuscript.
Author Response
Point 1. The authors gave good explanation to the reviewers but did not include much explanation in the manuscript. It is recommended that the authors include more discussion in the manuscript to improve the manuscript.
Response 1. Thank you for the helpful comment. Following your suggestion, we included more discussions in the revised manuscript in page 15 line 444-451 as follows.
As we enriched EpCAM-positive cells to establish new HOV cell lines, we cannot formally exclude the possibility that we excluded EpCAM-negative cancer cells and included normal epithelial cells. However, our every cohort showed a well-differentiated histology and we gained the similar histology in xenografts through our cell culture method. Moreover, we could also establish HOV26T sarcomatous cell line, which was probably EpCAM-negative, from sarcomatous component in the original tumor. Thus, these results lead us to conclude that our cell culture system is a superior method which is useful in terms of recapitulation of its original molecular and histological features faithfully.
Reviewer 3 Report
The authors replied to some of my concerns but to others
Point Point 5: In the opinion of this referee the recapitulation of immunohistochemical status of the new HOV cell lines has to be shown at first with the primary tumors and then contextualized in general in the immunohistochemical markers of ovarian carcinoma
Response 5: Thank you for the suggestion about the data presentation. It may be a good idea to demonstrate our main conclusion about the CDX model. However, we applied only several cell lines to the CDX model and as other reviewers do not suggest such change and we would like to state the results as they are.
Frankly this is not an explanation to say that the other reviews did not ask for it
Point 8: Cells transduced in vitro seem to growth very much when given DOX, while the cells not given DOX are not growing. This is strange as it was previously stated that cells lines were obtained and this implied that cells were growing. In any case the transduction of the vector could alter the cells and the xenografts derived from them. In addition the only histological examination (hematossilin and eosin) is too scanty to say that the growing tumor recapitulated the cells from which they derived. The positivity of a panel of mucionous proteins should be shown as compared to the original primary tumors.
Response 8: Thank you for the thoughtful comments. We agree to the reviewer’s comment that this looks strange as it was previously stated that cells lines were obtained and this implied that cells were growing. However, as the proliferate rate of HOV29T was exceptionally low and the actual doubling time in medium B was about 9 days, the cell growth in the presence of DOX was almost unchanged whereas the doubling time of the EMR-expressing HOV29 was about 28 hours indicating the transgene expression greatly enhanced its proliferation rate, which is indeed another merit of conditional expression of transgenes. To clarify this point, we modified a sentence in page 13 line 362 to 363 as follows.
We also agree to the reviewer’s comment that the transduction of the vector could alter the cells and the xenografts derived from them, and this could be a limitation of the current method. Finally, we agree to the reviewer’s critical comment that the only histological examination (hematoxilin and eosin) is too scanty to say that the growing tumor recapitulated the cells from which they derived and IHC analyses and other biological examination will be required to see whether the cell line recapitulate other features of original tumor cells which is an important scope of future study. However, we think this experiment showed a very strong impact that this DOX system restored not only the ability of mucin production but also the tumor histology itself in which all factors such as cell adhesion and/or cell structure were reflected. In that respect, we think that the most convincing and most impressive figure is to show the histology which is a culmination of all molecular pathological pathway and/or phenomena. Furthermore, to our knowledge, as xenograft models for mucinous ovarian carcinoma have not been recapitulated even at this level our method has still some advantage to make xenograft models from mucinous-type ovarian tumors.
I was asking to check by IHC in the mucinous transformed xenografts for some markers as was done for the other tumors, but the answer is that it will be the topic for other studies.
I do not think the authors have answered properly, and for me the paper does not deserve to be published unless new data are provided.
Author Response
Points 5:In the opinion of this referee the recapitulation of immunohistochemical status of the new HOV cell lines has to be shown at first with the primary tumors and then contextualized in general in the immunohistochemical markers of ovarian carcinoma
Response 5: Thank you for the suggestion about the data presentation. It may be a good idea to demonstrate our main conclusion about the CDX model. However, we applied only several cell lines to the CDX model and as other reviewers do not suggest such change and we would like to state the results as they are.
Point 5-2: Frankly this is not an explanation to say that the other reviews did not ask for it
Response 5-2: We are sorry that we used the other reviews as our explanation. We know there is a merit to show the recapitulation of immunohistochemical status of the new HOV cell lines be shown at first and that is one way of data presentation. However, as we applied only several cell lines to the CDX model, we would like to demonstrate the overall experimental structure first. We believe that the current sequence of the data presentation is easier for readers to follow. Needless to say, the significance of our study is unchanged with whatever order of data presentation.
Point 8: Cells transduced in vitro seem to growth very much when given DOX, while the cells not given DOX are not growing. This is strange as it was previously stated that cells lines were obtained and this implied that cells were growing. In any case the transduction of the vector could alter the cells and the xenografts derived from them. In addition the only histological examination (hematossilin and eosin) is too scanty to say that the growing tumor recapitulated the cells from which they derived. The positivity of a panel of mucionous proteins should be shown as compared to the original primary tumors.
Response 8: Thank you for the thoughtful comments. We agree to the reviewer’s comment that this looks strange as it was previously stated that cells lines were obtained and this implied that cells were growing. However, as the proliferate rate of HOV29T was exceptionally low and the actual doubling time in medium B was about 9 days, the cell growth in the presence of DOX was almost unchanged whereas the doubling time of the EMR-expressing HOV29 was about 28 hours indicating the transgene expression greatly enhanced its proliferation rate, which is indeed another merit of conditional expression of transgenes. To clarify this point, we modified a sentence in page 13 line 362 to 363 as follows.
We also agree to the reviewer’s comment that the transduction of the vector could alter the cells and the xenografts derived from them, and this could be a limitation of the current method. Finally, we agree to the reviewer’s critical comment that the only histological examination (hematoxilin and eosin) is too scanty to say that the growing tumor recapitulated the cells from which they derived and IHC analyses and other biological examination will be required to see whether the cell line recapitulate other features of original tumor cells which is an important scope of future study. However, we think this experiment showed a very strong impact that this DOX system restored not only the ability of mucin production but also the tumor histology itself in which all factors such as cell adhesion and/or cell structure were reflected. In that respect, we think that the most convincing and most impressive figure is to show the histology which is a culmination of all molecular pathological pathway and/or phenomena. Furthermore, to our knowledge, as xenograft models for mucinous ovarian carcinoma have not been recapitulated even at this level our method has still some advantage to make xenograft models from mucinous-type ovarian tumors.
Point 8-2: I was asking to check by IHC in the mucinous transformed xenografts for some markers as was done for the other tumors, but the answer is that it will be the topic for other studies. I do not think the authors have answered properly, and for me the paper does not deserve to be published unless new data are provided.
Response 8-2: We are sorry our response did not satisfy this reviewer. This reviewer recommend to add IHC data of mucin staining. However, the previous study examining expression profile of 36 mucins in ovarian mucinous tumors indicated that borderline and malignant ovarian mucinous tumors show similar mucin immune-profile being strongly and uniformly positive for MUC5AC (97.2% of cases), whereas only focally positive for MUC1 (19.4%), MUC2 (38.9%), and MUC6 (22.2%) (Int J Gynecol Pathol. 2014, 33:166-75). The authors concluded such IHC examination is useful to distinguish ovarian mucinous tumors from metastatic ovarian tumors from gastric cancer, pancreatic cancer and colon cancer.The results suggest that IHC data with a set of anti-mucin antibodies in our HOV xenografts and original tumors are not so informative to support our conclusion that our CDX model based on new culture methods can recapitulate original ovarian mucinous tumors as we excluded clinical samples of metastatic ovarian tumors and all the mucinous-type cases are ovarian mucinous tumors. Therefore, we do not think such IHC data as essential components in the whole context of our study. We can still observe mucin accumulation in the xenograft tumors as well as original tumors without such IHC data. Therefore, we responded that it is an important scope of future study. Furthermore, the editor gave us only 10 days for revision and it is impossible for us to revise the manuscript with such new data within 10 days.
We hope the editor and the reviewers understand the situation and make a fair decision on our revised manuscript.